# Beyond Sub-Gaussian Measurements:
# High-Dimensional Structured Estimation with
# Sub-Exponential Designs

**Vidyashankar Sivakumar**          **Arindam Banerjee**
Department of Computer Science & Engineering
University of Minnesota, Twin Cities
{sivakuma,banerjee}@cs.umn.edu

**Pradeep Ravikumar**
Department of Computer Science
University of Texas, Austin
pradeepr@cs.utexas.edu

## Abstract

We consider the problem of high-dimensional structured estimation with norm-regularized estimators, such as Lasso, when the design matrix and noise are drawn from sub-exponential distributions. Existing results only consider sub-Gaussian designs and noise, and both the sample complexity and non-asymptotic estimation error have been shown to depend on the Gaussian width of suitable sets. In contrast, for the sub-exponential setting, we show that the sample complexity and the estimation error will depend on the *exponential width* of the corresponding sets, and the analysis holds for any norm. Further, using generic chaining, we show that the exponential width for any set will be at most $\sqrt{\log p}$ times the Gaussian width of the set, yielding Gaussian width based results even for the sub-exponential case. Further, for certain popular estimators, viz Lasso and Group Lasso, using a VC-dimension based analysis, we show that the sample complexity will in fact be the same order as Gaussian designs. Our general analysis and results are the first in the sub-exponential setting, and are readily applicable to special sub-exponential families such as log-concave and extreme-value distributions.

## 1  Introduction

We consider the following problem of high dimensional linear regression:

$$y = X\theta^* + \omega \,, \tag{1}$$

where $y \in \mathbb{R}^n$ is the response vector, $X \in \mathbb{R}^{n \times p}$ has independent isotropic sub-exponential random rows, $\omega \in \mathbb{R}^n$ has i.i.d sub-exponential entries and the number of covariates $p$ is much larger compared to the number of samples $n$. Given $y$, $X$ and assuming that $\theta^*$ is 'structured', usually characterized as having a small value according to some norm $R(\cdot)$, the problem is to recover $\hat{\theta}$ close to $\theta^*$. Considerable progress has been made over the past decade on high-dimensional structured estimation using suitable M-estimators or norm-regularized regression [16, 2] of the form:

$$\hat{\theta}_{\lambda_n} = \operatorname*{argmin}_{\theta \in \mathbb{R}^p} \frac{1}{2n} \|y - X\theta\|_2^2 + \lambda_n R(\theta) \,, \tag{2}$$

where $R(\theta)$ is a suitable norm, and $\lambda_n > 0$ is the regularization parameter. Early work focused on high-dimensional estimation of sparse vectors using the Lasso and related estimators, where $R(\theta) = \|\theta\|_1$ [13, 22, 23]. Sample complexity of such estimators have been rigorously established based on the RIP(restricted isometry property) [4, 5] and the more general RE(restricted eigenvalue) conditions [3, 16, 2]. Several subsequent advances have considered structures beyond $\ell_1$, using more general norms such as (overlapping) group sparse norms, k-support norm, nuclear norm, and so on [16, 8, 7]. In recent years, much of the literature has been unified and nonasymptotic estimation error bound analysis techniques have been developed for regularized estimation with any norm [2].

In spite of such advances, most of the existing literature relies on the assumption that entries in the design matrix $X \in \mathbb{R}^{n \times p}$ are sub-Gaussian. In particular, recent unified treatments based on decomposable norms, atomic norms, or general norms all rely on concentration properties of sub-Gaussian distributions [16, 7, 2]. Certain estimators, such as the Dantzig selector and variants, consider a constrained problem rather than a regularized problem as in (2) but the analysis again relies on entries of $X$ being sub-Gaussian [6, 8]. For the setting of constrained estimation, building on prior work by [10], [20] outlines a possible strategy for such analysis which can work for any distribution, but works out details only for the sub-Gaussian case. In recent work [9] considered sub-Gaussian design matrices but with heavy-tailed noise, and suggested modifying the estimator in (1) via a median-of-means type estimator based on multiple estimates of $\hat{\theta}$ from sub-samples.

In this paper, we establish results for the norm-regularized estimation problem as in (2) for any norm $R(\theta)$ under the assumption that elements $X_{ij}$ of the design matrix $X \in \mathbb{R}^{n \times p}$ follow a sub-exponential distribution, whose tails are dominated by scaled versions of the (symmetric) exponential distribution, i.e., $P(|X_{ij}| > t) \leq c_1 \exp(-t/c_2)$ for all $t \geq 0$ and for suitable constants $c_1, c_2$ [12, 21]. To understand the motivation of our work, note that in most of machine learning and statistics, unlike in compressed sensing, the design matrix cannot be chosen but gets determined by the problem. In many application domains like finance, climate science, ecology, social network analysis, etc., variables with heavier tails than sub-Gaussians are frequently encountered. For example in climate science, to understand the relationship between extreme value phenomena like heavy precipitation variables from the extreme-value distributions are used. While high dimensional statistical techniques have been used in practice for such applications, currently lacking is the theoretical guarantees on their performance. Note that the class of sub-exponential distributions have heavier tails compared to sub-Gaussians but have all moments. To the best of our knowledge, this is the first paper to analyze regularized high-dimensional estimation problems of the form (2) with sub-exponential design matrices and noise.

In our main result, we obtain bounds on the estimation error $\|\hat{\Delta}_n\|_2 = \|\hat{\theta}_{\lambda_n} - \theta^*\|_2$, where $\theta^*$ is the optimal structured parameter. The sample complexity bounds are $\log p$ worse compared to the sub-Gaussian case. For example for the $\ell_1$ norm, we obtain $n = O(s \log^2 p)$ sample complexity bound instead of $O(s \log p)$ for the sub-Gaussian case. The analysis depends on two key ingredients which have been discussed in previous work [16, 2]: 1. The satisfaction of the RE condition on a set $A$ which is the error set associated with the norm, and 2. The design matrix-noise interaction manifested in the form of lower bounds on the regularization parameter. Specifically, the RE condition depends on the properties of the design matrix. We outline two different approaches for obtaining the sample complexity, to satisfy the RE condition: one based on the 'exponential width' of $A$ and another based on the VC-dimension of linear predictors drawn from $A$ [10, 20, 11]. For two widely used cases, Lasso and group-lasso, we show that the VC-dimension based analysis leads to a sharp bound on the sample complexity, which is exactly the same order as that for sub-Gaussian design matrices! In particular, for Lasso with $s$-sparsity, $O(s \log p)$ samples are sufficient to satisfy the RE condition for sub-exponential designs. Further, we show that the bound on the regularization parameter depends on the 'exponential width' $w_e(\Omega_R)$ of the unit norm ball $\Omega_R = \{u \in \mathbb{R}^p | R(u) \leq 1\}$. Through a careful argument based on generic chaining [19], we show that for any set $T \subset \mathbb{R}^p$, the exponential width $w_e(T) \leq c w_g(T) \sqrt{\log p}$, where $w_g(T)$ is the Gaussian width of the set $T$ and $c$ is an absolute constant. Recent advances on computing or bounding $w_g(T)$ for various structured sets can then be used to bound $w_e(T)$. Again, for the case of Lasso, $w_e(\Omega_R) \leq c \log p$.

The rest of the paper is organized as follows. In Section 2 we describe various aspects of the problem and highlight our contributions. In Section 3 we establish a key result on the relationship between Gaussian and exponential widths of sets which will be used for our subsequent analysis. In Section 4 we establish results on the regularization parameter $\lambda_n$, RE constant $\kappa$ and the non-asymptotic estimation error $\|\hat{\Delta}_n\|_2$. We show some experimental results before concluding in Section 6.

## 2 Background and Preliminaries

In this section, we describe various aspects of the problem, introducing notations along the way, and highlight our contributions. Throughout the paper values of constants change from line to line.

## 2.1 Problem setup

We consider the problem defined in (2). The goal of this paper is to establish conditions for consistent estimation and derive bounds on $\|\hat{\Delta}_n\|_2 = \|\hat{\theta} - \theta^*\|_2$.

**Error set:** Under the assumption $\lambda_n \geq \beta R^*(\frac{1}{n}X^T(y - X\theta^*)), \beta > 1$, the error vector $\hat{\Delta}_n = \hat{\theta} - \theta^*$ lies in a cone $A \subseteq S^{p-1}$ [3, 16, 2].

**Regularization parameter:** For $\beta > 1$, $\lambda_n \geq \beta R^*(\frac{1}{n}X^T(y - X\theta))$ following analysis in [16, 2].

**Restricted Eigenvalue (RE) conditions:** For consistent estimation, the design matrix $X$ should satisfy the following RE condition $\inf_{u \in A} \frac{1}{\sqrt{n}}\|Xu\|_2 \geq \kappa$ on the error set $A$ for some constant $\kappa > 0$ [3, 16, 2, 20, 18]. The RE sample complexity is the number of samples $n$ required to satisfy the RE condition and has been shown to be related to the Gaussian width of the error set. [7, 2, 20].

**Deterministic recovery bounds:** If $X$ satisfies the RE condition on the error set $A$ and $\lambda_n$ satisfies the assumptions stated earlier, [2] show the error bound $\|\hat{\Delta}_n\|_2 \leq c\Psi(A)\frac{\lambda_n}{\kappa}$ with high probability (w.h.p), for some constant $c$, where $\Psi(A) = \sup_{u \in A} \frac{R(u)}{\|u\|_2}$ is the norm compatibility constant.

**$\ell_1$ norm regularization:** One example for $R(\cdot)$ we will consider throughout the paper is the $\ell_1$ norm regularization. In particular we will always consider $\|\theta^*\|_0 = s$.

**Group-sparse norms:** Another popular example we consider is the group-sparse norm. Let $\mathcal{G} = \{\mathcal{G}_1, \mathcal{G}_2, \ldots, \mathcal{G}_{N_{\mathcal{G}}}\}$ denote a collection of groups, which are blocks of any vector $\theta \in \mathbb{R}^p$. For any vector $\theta \in \mathbb{R}^p$, let $\theta^{N_{\mathcal{G}}}$ denote a vector with coordinates $\theta_i^{N_{\mathcal{G}}} = \theta_i$ if $i \in \mathcal{G}_{N_{\mathcal{G}}}$, else $\theta_i^{N_{\mathcal{G}}} = 0$. Let $m = \max_{i \in [1, \cdots, N_{\mathcal{G}}]} |\mathcal{G}_i|$ be the maximum size of any group. In the group sparse setting for any subset $S_{\mathcal{G}} \subseteq \{1, 2, \ldots, N_{\mathcal{G}}\}$ with cardinality $|S_{\mathcal{G}}| = s_{\mathcal{G}}$, we assume that the parameter vector $\theta^* \in \mathbb{R}^p$ satisfies $\theta^{*N_{\mathcal{G}}} = \vec{0}, \forall N_{\mathcal{G}} \notin S_{\mathcal{G}}$. Such a vector is called $S_{\mathcal{G}}$-group sparse. We will focus on the case when $R(\theta) = \sum_{i=1}^{N_{\mathcal{G}}} \|\theta^i\|_2$.

## 2.2 Contributions

One of our major results is the relationship between the Gaussian and exponential width of sets using arguments from generic chaining [19]. Existing analysis frameworks for our problem for sub-Gaussian $X$ and $\omega$ obtain results in terms of Gaussian widths of suitable sets associated with the norm [2, 20]. For sub-exponential $X$ and $\omega$ this dependency, in some cases, is replaced by the exponential width of the set. By establishing a precise relationship between the two quantities, we leverage existing results on the computation of Gaussian widths for our scenario. Another contribution is obtaining the same order of the RE sample complexity bound as for the sub-Gaussian case for $\ell_1$ and group-sparse norms. While this strong result has already been explored in [11] for $\ell_1$, we adapt it for our analysis framework and also extend it to the group-sparse setting. As for the application of our work, the results are applicable to all log-concave distributions which by definition are distributions admitting a log-concave density $f$ i.e. a density of the form $f = e^\Psi$ with $\Psi$ any concave function. This covers many practically used distributions including extreme value distributions.

## 3 Relationship between Gaussian and Exponential Widths

In this section we introduce a complexity parameter of a set $w_e(\cdot)$, which we call the exponential width of the set, and establish a sharp upper bound for it in terms of the Gaussian width of the set $w_g(\cdot)$. In particular, we prove the inequality: $w_e(A) \leq c \cdot w_g(A)\sqrt{\log p}$ for some fixed constant $c$. To see the connection with the rest of the paper, remember that our subsequent results for $\lambda_n$ and $\kappa$ are expressed in terms of the Gaussian width and exponential width of specific sets associated with the norm. With this result, we establish precise sample complexity bounds by leveraging a body of literature on the computation of Gaussian widths for various structured sets [7, 20]. We note that while the exponential width has been defined and used earlier, see for e.g. [19, 15], to the best of our knowledge this is the first result establishing the relation between the Gaussian and exponential widths of sets. Our result relies on generic chaining [19].

## 3.1 Generic Chaining, Gaussian Width and Exponential Widths

Consider a process $\{X_t\}_{t \in T} = \langle h, t \rangle$ indexed by a set $T \subseteq \mathbb{R}^p$, where each element $h_i$ has mean 0. It follows from the definition that the process is centered, i.e., $E(X_t) = 0, \forall t \in T$. We will also assume for convenience w.l.o.g that set $T$ is finite. Also, for any $s, t \in T$, consider a canonical distance metric $d(s, t)$. We are interested in computing the quantity $E \sup_{t \in T} X_t$. Now, for reasons detailed in the supplement, consider that we split $T$ into a sequence of subsets $T_0 \subseteq T_1 \subseteq \ldots \subseteq T$, with $T_0 = \{t_0\}$, $|T_n| \leq 2^{2^n}$ for $n \geq 1$ and $T_m = T$ for some large $m$. Let function $\pi_n : T \to T_n$, defined as $\pi_n(t) = \{s : d(s, t) \leq d(s_1, t), \forall s, s_1 \in T_n\}$, maps each point $t \in T$ to some point $s \in T_n$ closest according to $d$. The set $T_n$ and the associated function $\pi_n$ define a partition $\mathcal{A}_n$ of the set $T$. Each element of the partition $\mathcal{A}_n$ has some element $s \in T_n$ and all $t \in T$ closest to it according to the map $\pi_n$. Also the size of the partition $|\mathcal{A}_n| \leq 2^{2^n}$. $\mathcal{A}_n$ are called admissible sequences in generic chaining. Note that there are multiple admissible sequences corresponding to multiple ways of defining the sets $T_0, T_1, \ldots, T_m$. We will denote by $\Delta(A_n(t))$ the diameter of the element $A_n(t)$ w.r.t distance metric $d$ defined as $\Delta(A_n(t)) = \sup_{s, t \in A_n(t)} d(s, t)$.

**Definition 1 $\gamma$-functionals:** [19] Given $\alpha > 0$, and a metric space $(T, d)$ we define

$$\gamma_\alpha(T, d) = \inf \sup_t \sum_{n \geq 0} 2^{n/\alpha} \Delta(A_n(t)) \,, \tag{3}$$

where the $\inf$ is taken over all possible admissible sequences of the set $T$.

**Gaussian width:** Let $\{X_t\}_{t \in T} = \langle g, t \rangle$ where each element $g_i$ is i.i.d $N(0, 1)$. The quantity $w_g(T) = E \sup_{t \in T} X_t$ is called the Gaussian width of the set $T$. Define the distance metric $d_2(s, t) = \|s - t\|_2$. The relation between Gaussian width and the $\gamma$-functionals is seen from the following result from [Theorem 2.1.1] of [19] stated below:

$$\frac{1}{L} \gamma_2(T, d_2) \leq w_g(T) \leq L \gamma_2(T, d_2) \,. \tag{4}$$

Note that, following [Theorem 2.1.5] in [19] any process which satisfies the concentration bound $P(|X_s - X_t| \geq u) \leq 2 \exp\left(-\frac{u^2}{d_2(s,t)^2}\right)$ satisfies the upper bound in (4).

**Exponential width:** Let $\{X_t\}_{t \in T} = \langle e, t \rangle$ where each element $e_i$ is is a centered i.i.d exponential random variable satisfying $P(|e_i| \geq u) = \exp(-u)$. Define the distance metrics $d_2(s, t) = \|s - t\|_2$ and $d_\infty(s, t) = \|s - t\|_\infty$. The quantity $w_e(T) = E \sup_{t \in T} X_t$ is called the exponential width of the set $T$. By [Theorem 1.2.7] and [Theorem 5.2.7] in [19], for some universal constant $L$, $w_e(T)$ satisfies:

$$\frac{1}{L}(\gamma_2(T, d_2) + \gamma_1(T, d_\infty)) \leq w_e(T) \leq L(\gamma_2(T, d_2) + \gamma_1(T, d_\infty)) \tag{5}$$

Note that any process which satisfies the sub-exponential concentration bound $P(|X_s - X_t| \geq u) \leq 2 \exp\left(-K \min\left(\frac{u^2}{d_2(s,t)^2}, \frac{u}{d_\infty(s,t)}\right)\right)$ satisfies the upper bound in the above inequality [15, 19].

## 3.2 An Upper Bound for the Exponential Width

In this section we prove the following relationship between the exponential and Gaussian widths:

**Theorem 1** *For any set $T \subset \mathbb{R}^p$, for some constant $c$ the following holds:*

$$w_e(T) \leq c \cdot w_g(T) \sqrt{\log p} \,. \tag{6}$$

*Proof:* The result depends on geometric results [Lemma 2.6.1] and [Theorem 2.6.2] in [19].

**Theorem 2** *[19] Consider a countable set $T \subset \mathbb{R}^p$, and a number $u > 0$. Assume that the Gaussian width is bounded i.e. $S = \gamma_2(T, d_2) \leq \infty$. Then there is a decomposition $T \subset T_1 + T_2$ where $T_1 + T_2 = \{t_1 + t_2 : t_1 \in T_1, t_2 \in T_2\}$, such that*

$$\gamma_2(T_1, d_2) \leq LS \,, \qquad \gamma_1(T_1, d_\infty) \leq LSu \tag{7}$$

$$\gamma_2(T_2, d_2) \leq LS \,, \qquad T_2 \subset \frac{LS}{u} B_1 \,, \tag{8}$$

*where $L$ is some universal constant and $B_1$ is the unit $\ell_1$ norm ball in $\mathbb{R}^p$.*

We first examine the exponential widths of the sets $T_1$ and $T_2$. For the set $T_1$:

$$w_e(T_1) \leq L[\gamma_2(T_1, d_2) + \gamma_1(T_1, d_\infty)] \leq L[S + Su] = L(w_g(T) + w_g(T)u) , \qquad (9)$$

where the first inequality follows from (5) and the second inequality follows from (7). We will need the following result on bounding the exponential width of an unit $\ell_1$-norm ball in $p$ dimensions to compute the exponential width of $T_2$. The proof, given in the supplement, is based on the fact $\sup_{t \in B_1} \langle e, t \rangle = \|e\|_\infty$ and then using a simple union bound argument to bound $\|e\|_\infty$.

**Lemma 1** *Consider the set $B_1 = \{t \in \mathbb{R}^p : \|t\|_1 \leq 1\}$. Then for some universal constant $L$:*

$$w_e(B_1) = E\left[\sup_{t \in B_1} \langle e, t \rangle\right] \leq L \log p . \qquad (10)$$

The exponential width of $T_2$ is:

$$w_e(T_2) = w_e((LS/u)B_1) = (LS/u)w_e(B_1) = (L/u)w_g(T)w_e(B_1) \leq (L/u)w_g(T)\log p . \quad (11)$$

The first equality follows from (8) as $T_2$ is a subset of a $(LS/u)$-scaled $\ell_1$ norm ball, the second inequality follows from elementary properties of widths of sets and the last inequality follows from Lemma 1. Now as stated in Theorem 2, $u$ in (9) and (11) is any number greater than 0. We choose $u = \sqrt{\log p}$ and noting that $(1 + \sqrt{\log p}) \leq L\sqrt{\log p}$ for some constant $L$ yields:

$$w_e(T_1) \leq Lw_g(T)\sqrt{\log p}, \qquad w_e(T_2) \leq Lw_g(T)\sqrt{\log p} \qquad (12)$$

The final step, following arguments as [Theorem 2.1.6] [19], is to bound exponential width of set $T$.

$$w_e(T) = E[\sup_{t \in T} \langle h, t \rangle] \leq E[\sup_{t_1 \in T_1} \langle h, t_1 \rangle] + E[\sup_{t_2 \in T_2} \langle h, t_2 \rangle] \leq w_e(T_1) + w_e(T_2) \leq Lw_g(T)\sqrt{\log p} .$$

This proves Theorem 1. ∎

## 4 Recovery Bounds

We obtain bounds on the error vector $\hat{\Delta}_n = \hat{\theta} - \theta^*$. If the regularization parameter $\lambda_n \geq \beta R^*(\frac{1}{n}X^T(y - X\theta^*)), \beta > 1$ and the RE condition is satisfied on the error set $A$ with RE constant $\kappa$, then [2, 16] obtain the following error bound w.h.p for some constant $c$:

$$\|\hat{\Delta}_n\|_2 \leq c \cdot \frac{\lambda_n}{\kappa}\Psi(A) , \qquad (13)$$

where $\Psi(A)$ is the norm compatibility constant given by $\sup_{u \in A}(R(u)/\|u\|_2)$.

### 4.1 Regularization Parameter

As discussed earlier, for our analysis the regularization parameter should satisfy $\lambda_n \geq \beta R^*(\frac{1}{n}X^T(y - X\theta^*)), \beta > 1$. Observe that for the linear model (1), $\omega = y - X\theta^*$ is the noise, implying that $\lambda_n \geq \beta R^*(\frac{1}{n}X^T\omega)$. With $e$ denoting a sub-exponential random vector with i.i.d entries,

$$E\left[R^*\left(\frac{1}{n}X^T\omega\right)\right] = E\left[\sup_{u \in \Omega_R} \|\omega\|_2 \left\langle \frac{1}{n}X^T\frac{\omega}{\|\omega\|_2}, u \right\rangle\right] = \frac{1}{n}E[\|\omega\|_2]E\left[\sup_{u \in \Omega_R} \langle e, u \rangle\right] . \quad (14)$$

The first equality follows from the definition of dual norm. The second inequality follows from the fact that $X$ and $\omega$ are independent of each other. Also by elementary arguments [21], $e = X^T(\omega/\|\omega\|_2)$ has i.i.d sub-exponential entries with sub-exponential norm bounded by $\sup_{\omega \in \mathbb{R}^n} \|\langle X_i^T, \omega/\|\omega\|_2 \rangle\|_{\psi_1}$. The above argument was first proposed for the sub-Gaussian case in [2]. For sub-exponential design and noise, the difference compared to the sub-Gaussian case is the dependence on the exponential width instead of the Gaussian width of the unit norm ball. Using known results on the Gaussian widths of unit $\ell_1$ and group-sparse norms, corollaries below are derived using the relationship between Gaussian and exponential widths derived in Section 3:

**Corollary 1** *If $R(\cdot)$ is the $\ell_1$ norm, for sub-exponential design matrix $X$ and noise $\omega$,*

$$E\left[R^*\left(\frac{1}{n}X^T(y - X\theta^*)\right)\right] \leq \frac{\eta_0}{\sqrt{n}}\log p \,. \tag{15}$$

**Corollary 2** *If $R(\cdot)$ is the group-sparse norm, for sub-exponential design matrix $X$ and noise $\omega$,*

$$E\left[R^*\left(\frac{1}{n}X^T(y - X\theta^*)\right)\right] \leq \frac{\eta_0}{\sqrt{n}}\sqrt{(m + \log N_{\mathcal{G}})\log p} \,. \tag{16}$$

## 4.2 The RE condition

For Gaussian and sub-Gaussian $X$, previous work has established RIP bounds of the form $\kappa_1 \leq \inf_{u \in A}(\frac{1}{\sqrt{n}})\|Xu\|_2 \leq \sup_{u \in A}(\frac{1}{\sqrt{n}})\|Xu\|_2 \leq \kappa_2$. In particular, RIP is satisfied w.h.p if the number of samples is of the order of square of the Gaussian width of the error set ,i.e., $O(w_g^2(A))$, which we will call the sub-Gaussian RE sample complexity bound. As we move to heavier tails, establishing such two-sided bounds requires assumptions on the boundedness of the Euclidean norm of the rows of $X$ [15, 17, 10]. On the other hand, analysis of only the lower bound requires very few assumptions on $X$. In particular, $\|Xu\|_2$ being the sum of random non-negative quantities the lower bound should be satisfied even with very weak moment assumptions on $X$. Making these observations, [10, 17] develop arguments obtaining sub-Gaussian RE sample complexity bounds when set $A$ is the unit sphere $S^{p-1}$ even for design matrices having only bounded fourth moments. Note that with such weak moment assumptions, a non-trivial non-asymptotic upper bound cannot be established. Our analysis for the RE condition essentially follow this premise and arguments from [10].

### 4.2.1 A Bound Based on Exponential Width

We obtain a sample complexity bound which depends on the exponential width of the error set $A$. The result we state below follows along similar arguments made in [20], which in turn are based on arguments from [10, 14].

**Theorem 3** *Let $X \in \mathbb{R}^{n \times p}$ have independent isotropic sub-exponential rows. Let $A \subseteq S^{p-1}$, $0 < \xi < 1$, and $c$ is a constant that depends on the sub-exponential norm $K = \sup_{u \in A}\||\langle X, u\rangle|\|_{\psi_1}$. Let $w_e(A)$ denote the exponential width of the set. Then for some $\tau > 0$ with probability atleast $(1 - \exp(-\tau^2/2))$,*

$$\inf_{u \in A}\|Xu\|_2 \geq c\xi(1 - \xi^2)^2\sqrt{n} - 4w_e(A) - \xi\tau \,. \tag{17}$$

Contrasting the result (17) with previous results for the sub-Gaussian case [2, 20] the dependence on $w_g(A)$ on the r.h.s is replaced by $w_e(A)$, thus leading to a $\log p$ worse sample complexity bound. The corollary below applies the result for the $\ell_1$ norm. Note that results from [1] for $\ell_1$ norm show RIP bounds w.h.p for the same number of samples.

**Corollary 3** *For an s-sparse $\theta^*$ and $\ell_1$ norm regularization, if $n \geq c \cdot s\log^2 p$ then with probability atleast $(1 - \exp(-\tau^2/2))$ and constants $c, \kappa$ depending on $\xi$ and $\tau$,*

$$\inf_{u \in A}\|Xu\|_2 \geq \kappa \,. \tag{18}$$

### 4.2.2 A Bound Based on VC-Dimensions

In this section, we show a stronger sub-Gaussian RE sample complexity result for sub-exponential $X$ and $\ell_1$, group-sparse regularization. The arguments follow along similar lines to [11, 10].

**Theorem 4** *Let $X \in \mathbb{R}^{n \times p}$ be a random matrix with isotropic random sub-exponential rows $X_i \in \mathbb{R}^p$. Let $A \subseteq S^{p-1}$, $0 < \xi < 1$, $c$ is a constant that depends on the sub-exponential norm $K = \sup_{u \in A}\||\langle X, u\rangle|\|_{\psi_1}$ and define $\beta = c(1 - \xi^2)^2$. Let $w_e(A)$ denote the exponential width of the set*

*A. Let $C_\xi = \{\mathbb{I}[|\langle X_i, u \rangle| > \xi], u \in A\}$ be a VC-class with VC-dimension $VC(C_\xi) \leq d$. For some suitable constant $c_1$, if $n \geq c_1(d/\beta^2)$, then with probability atleast $1 - \exp(-\eta_0 \beta^2 n)$:*

$$\inf_{u \in A} \frac{1}{\sqrt{n}} \|Xu\|_2 \geq \frac{c\xi(1 - \xi^2)^2}{2} \ . \tag{19}$$

Consider the case of $\ell_1$ norm. A consequence of the above result is that the RE condition is satisfied on the set $B = \{u|\|u\|_0 = s_1\} \cap S^{p-1}$ for some $s_1 \geq c \cdot s$ where $c$ is a constant that will depend on the RE constant $\kappa$ when $n$ is $O(s_1 \log p)$. The argument follows from the fact that $B \cap S^{p-1}$ is a union of $\binom{p}{s_1}$ spheres. Thus the result is obtained by applying Theorem 4 to each sphere and using a union bound argument. The final step involves showing that the RE condition is satisfied on the error set $A$ if it is satisfied on $B$ using Maurey's empirical approximation argument [17, 18, 11].

**Corollary 4** *For set $A \subseteq S^{p-1}$, which is the error set for the $\ell_1$ norm, if $n \geq c_2 s \log(ep/s)/\beta^2$ for some suitable constant $c_2$, then with probability atleast $1 - \exp(-\eta_0 n \beta^2) - \frac{1}{w^{\eta_1} p^{\eta_1 - 1}}$, where $\eta_0, \eta_1, w > 1$ are constants, the following result holds for $\kappa$ depending on the constant $\xi$:*

$$\inf_{u \in A} \frac{1}{\sqrt{n}} \|Xu\|_2 \geq \kappa \ . \tag{20}$$

Essentially the same arguments for the group-sparse norm lead to the following result:

**Corollary 5** *For set $A \subseteq S^{p-1}$, which is the error set for the group-sparse norm, if $n \geq (c(ms_\mathcal{G} + s_\mathcal{G} \log(eN_\mathcal{G}/s_\mathcal{G})))/\beta^2$, then with probability atleast $1 - \exp(-\eta_0 n \beta^2) - \frac{1}{w^{\eta_1} N_\mathcal{G}^{\eta_1 - 1} m^{\eta_1 - 1}}$ where $\eta_0, \eta_1, w > 1$ are constants and $\kappa$ depending on constant $\xi$,*

$$\inf_{u \in A} \frac{1}{\sqrt{n}} \|Xu\|_2 \geq \kappa \ . \tag{21}$$

### 4.3 Recovery Bounds for $\ell_1$ and Group-Sparse Norms

We combine result (13) with results obtained for $\lambda_n$ and $\kappa$ previously for $\ell_1$ and group-sparse norms.

**Corollary 6** *For the $\ell_1$ norm, when $n \geq cs \log p$ for some constant $c$, with high probability:*

$$\|\hat{\Delta}_n\|_2 \leq O(\sqrt{s} \log p / \sqrt{n}) \ . \tag{22}$$

**Corollary 7** *For the group-sparse norm, when $n \geq c(ms_\mathcal{G} + s_\mathcal{G} \log(N_\mathcal{G}))$, for some constant $c$, with high probability:*

$$\|\hat{\Delta}_n\|_2 \leq O\left(\sqrt{\frac{s_\mathcal{G} \log p(m + \log N_\mathcal{G})}{n}}\right) \ . \tag{23}$$

Both bounds are $\sqrt{\log p}$ worse compared to corresponding bounds for the sub-Gaussian case. In terms of sample complexity, $n$ should scale as $O(s \log^2 p)$, instead of $O(s \log p)$ for sub-Gaussian, for $\ell_1$ norm and $O(s_\mathcal{G} \log p(m + \log N_\mathcal{G}))$, instead of $O(s_\mathcal{G}(m + \log N_\mathcal{G}))$ for the sub-Gaussian case, for group-sparse lasso to get upto a constant order error bound.

## 5 Experiments

We perform experiments on synthetic data to compare estimation errors for Gaussian and sub-exponential design matrices and noise for both $\ell_1$ and group sparse norms. For $\ell_1$ we run experiments with dimensionality $p = 300$ and sparsity level $s = 10$. For group sparse norms we run experiments with dimensionality $p = 300$, max. group size $m = 6$, number of groups $N_\mathcal{G} = 50$ groups each of size 6 and 4 non-zero groups. For the design matrix $X$, for the Gaussian case we sample rows randomly from an isotropic Gaussian distribution, while for sub-exponential design

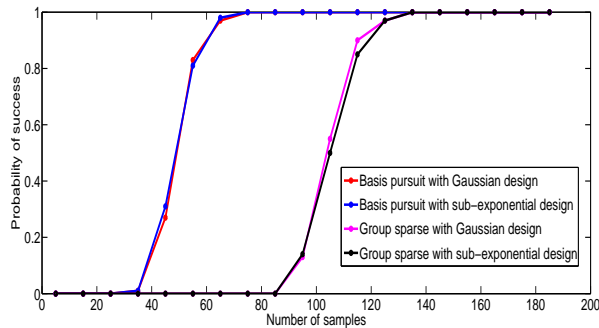

*Figure 1:* Probability of recovery in noiseless case with increasing sample size. There is a sharp phase transition and the curves overlap for Gaussian and sub-exponential designs.

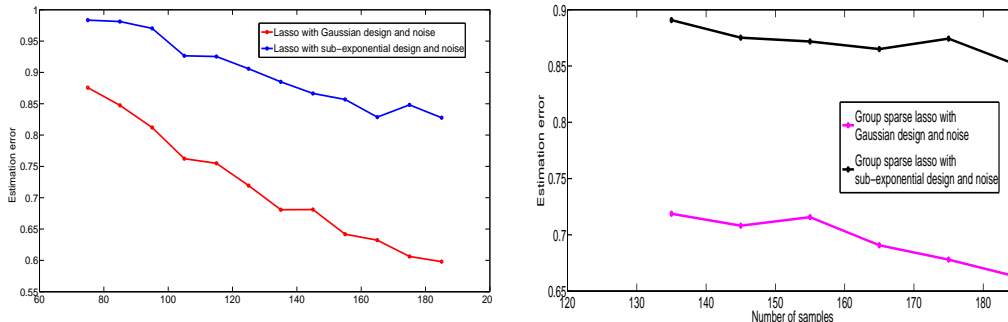

*Figure 2:* Estimation error $\|\hat{\Delta}_n\|_2$ vs sample size for $\ell_1$ (left) and group-sparse norms (right). The curve for sub-exponential designs and noise decays slower than Gaussians.

matrices we sample each row of $X$ randomly from an isotropic extreme-value distribution. The number of samples $n$ in $X$ is incremented in steps of 10 with an initial starting value of 5. For the noise $\omega$, it is sampled i.i.d from the Gaussian and extreme-value distributions with variance 1 for the Gaussian and sub-exponential cases respectively. For each sample size $n$, we repeat the procedure above 100 times and all results reported in the plots are average values over the 100 runs. We report two sets of results. Figure 1 shows percentage of success vs sample size for the noiseless case when $y = X\theta^*$. A success in the noiseless case denotes exact recovery which is possible when the RE condition is satisfied. Hence we expect the sample complexity for recovery to be order of square of Gaussian width for Gaussian and extreme-value distributions as validated by the plots in Figure 1. Figure 2 shows average estimation error vs number of samples for the noisy case when $y = X\theta^* + \omega$. The noise is added only for runs in which exact recovery was possible in the noiseless case. For example when $n = 5$ we do not have any results in Figure 2 as even noiseless recovery is not possible. For each $n$, the estimation errors are average values over 100 runs. As seen in Figure 2, the error decay is slower for extreme-value distributions compared to the Gaussian case.

## 6   Conclusions

This paper presents a unified framework for analysis of non-asymptotic error and structured recovery in norm regularized regression problems when the design matrix and noise are sub-exponential, essentially generalizing the corresponding analysis and results for the sub-Gaussian case. The main observation is that the dependence on Gaussian width is replaced by the exponential width of suitable sets associated with the norm. Together with the result on the relationship between exponential and Gaussian widths, previous analysis techniques essentially carry over to the sub-exponential case. We also show that a stronger result exists for the RE condition for the Lasso and group-lasso problems. As future work we will consider extending the stronger result for the RE condition for all norms.

**Acknowledgements:**  This work was supported by NSF grants IIS-1447566, IIS-1447574, IIS-1422557, CCF-1451986, CNS-1314560, IIS-0953274, IIS-1029711, and by NASA grant NNX12AQ39A.

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
