[Reviews · NeurIPS 2015]

Submitted by Assigned_Reviewer_1

The paper discusses high-dimensional sparse/group sparse estimation with l1/l1-l2 regularized least squares estimation. As opposed to the usual setting with Gaussian measurements and noise, the author consider isotropic sub-exponential measurements and i.i.d. subexponential noise. The main result of the paper is that the error/sample complexities bounds are essentially the same as for the usual setting, modulo an extra log factor. The results are based on advanced techniques from empirical processes.

In that analysis, the extra log factor results from the transition to the Gaussian width (usual setting) to the exponential width. Altogether the results can be seen as a meaningful extension of earlier results (Negahban et al[18], Rudelson & Zhou [20], and others).

Quality/Clarity: The technical quality of the paper is consistently high. It is well-written and organized.

Originality/Significance: The paper studies a domain that has been extensively studied in the past years. The contribution of the paper is significant for experts in that domain, but would probably be considered as minor for the general audience. It is also not clear to what extent the techniques are novel given a series of related works on compressed sensing with measurements having heavier tails.

Specific comments/questions: - the authors study only isotropic measurements, wheres the non-isotropic sub-Gaussian case has been dealt with successfully in Rudelson and Zhou [20].

Maybe the authors can comment on how their results could be extended in this direction - Gaussian-width based analysis is used excessively in Vershynin's tutorial on high-dimensional estimation. In my view, it is helpful to that cite that work - the experiments are particularly disappointing as there is no comparison between sub-Gaussian and sub-Exponential measurements. Such comparison would allow one to assess the sharpness of the authors' main result.
Summary: A valuable paper for people interested in the theory of compressed sensing and high-dimensional statistics.

Submitted by Assigned_Reviewer_2

Few typos (I include things to be added inside []): Line 321,330,344,358,368,: "at[ ]least"

Summary: While the theoretical sub-community would appreciate the technical complexity of going from sub-Gaussian to sub-exponential designs, I believe the paper is weak on motivation. Independently of this, the paper has good theoretical results and experimental setup.

Submitted by Assigned_Reviewer_3

This paper consider the high-dimensional estimation problems with sub-exponential design and noise. The authors establish estimation error bound via the exponential width argument, show that the estimation error will be at most \sqrt{\log p} times worse than the case for sub-Gaussian.

The first result is the connection between Gaussian and exponential widths. It is shown that the exponential widths is upper bounded by Gaussian widths by a fact of c \sqrt{\log p}.

The second important result shows to obtain restricted eigenvalue condition, for sub-exponential design the sample complexity is the same as sub-Gaussian case. The authors outlined two prooof techniques and shows that the bound based on exponential widths argument is \log p worse than the one obtained by VC dimension.

The authors also conducted some simulations to verify the proved sample complexity. However, I don't think the way Figure 1 presented is very informative: to show the sample complexity is O(s \log^2 p) instead of O(s \log p), its better to plot the normalized sample size (n/s \log^2 p) versus the probability of success to see whether the curves for different dimensions are close enough.

Overall, I think this paper established some substantial results for high dimensional estimation under exponential design and noise. The proof technique is novel and interesting as well. I would like to see it appear in NIPS.
Summary: This paper establish the high-dimensional estimation error bound with sub-exponential design and noise, interesting and useful analysis were presented.

Submitted by Assigned_Reviewer_4

The author considers the regression problem, where one is given

$$ y = X \theta^* + \omega $$

for some design matrix $X$ and noise vector $\omega$. The goal is to recover some estimate $\hat{\theta}$ of $\theta^*$ such that $\|\hat{theta} - \theta^*\|_2$ is small with high probability (over the noise vector $\omega$ as well as randomness in the design matrix $X$). The author considers the standard approach of recovering $\hat{\theta}$ as the solution to an optimization problem (a norm-regularized least squares regression).

Much attention has been given to this problem in the past when $X$ and $\omega$ have subgaussian entries. In that case, the number of rows $n$ of the design matrix $X$ that is needed relates to the gaussian mean width of the "error set" $A$ defined in lines 108-113 of the paper. In this submission, attention is given to the case when "subgaussian" is replaced with "subexponential". In this case, $n$ relates to the exponential width of $A$. A theorem is given (Theorem 1) stating that exponential width is at most gaussian width times $\sqrt{\log p}$ for any subset T of $\mathbb{R}^p$. This theorem is a simple corollary of two theorems from Talagrand's book after computing the exponential width of the unit $\ell_1$ ball, which is very standard (Theorems 1.2.7 and 2.6.2 in that book; Theorem 5.2.7 is also cited in the submission, but it is irrelevant, it provides the lower bound on the exponential width in terms of the $\gamma$-functionals, but that lower bound is never used in any of the proofs in the submission, and furthermore the lower bound is only true anyway when the r.v.'s *are* exponential and not merely subexponential, unlike this submission).

Technically, I found the jump from analyzing the gaussian case in previous work to the subexponential case in this submission to be quite a small jump. The new main theorem, Theorem 1, follows essentially immediately from the two theorems cited in Talagrand's book.

In any case, not having a big technical contribution is perfectly OK. More importantly, I think the submission could benefit greatly from more time spent on motivation in the introduction. My perhaps wrong impression is that one can sometimes choose their design matrix, and in those cases they can choose their entries to be subgaussian to get away with few measurements (depending on the gaussian width of the error set). In these cases where a choice is present, it seems there is no motivation to choose to use a design matrix with subexponential entries since the width parameter only gets worse. The question then is: are there strong motivating examples when you cannot choose your design matrix (it is simply given to you by the real world) *AND* it makes sense to model the entries in that design matrix as being subexponential? As far as I can see, the main motivation for this submission hinges upon a positive answer to this question, yet this question does not seem to be addressed adequately in the introduction. (The answer may very well be yes, but I have no idea, and the author should spend considerable time on concrete such examples in the introduction if they exist, since it justifies the importance of the whole paper.)

OTHER COMMENTS:

FROM THE MAIN PAPER: * line 070: "Note that sub-exponentials are the class of distributions which have heavier tails compared to sub-Gaussians and for which all moments exist.". I don't agree with the word "the" here before "class", since it is one of many. The distribution with pdf e^{-|x|^{1.5}} has heavier tails than sub-gaussians with all moments existing also.

* line 071: "Distributions having heavier tails than sub-exponentials start losing moments." This doesn't sound right to me. What about a distribution with pdf e^{-sqrt(|x|)}? The integral of x^p e^{-sqrt(x)} from 0 to infinity converges for any constant p greater than 0 (i.e. all moments exist). * Lines 084 and 085, $A$ is used without yet being defined and is thus somewhat confusing. It is only clear later in line 109 what $A$ was referring to. * I'm confused by line 115 which presents an inequality for $\lambda_n$. $\lambda_n$ is something you set in (2) so that the solution to the optimization problem has some desired property (e.g. it's close to $\theta^*$ in some norm with high probability). Are you saying it should be set to satisfy the given inequality? And are you saying there's a *specific* $\beta$ bigger than 1 for which it should be set this way? I cannot understand what this line means as written (although I guessed the earlier part of this bullet based on line 108). * Lines 199-201, it states (4) holds for any g whose entries have subgaussian decay. This is false. The upper bound of (4) is indeed true as long as there is subgaussian decay, but the lower bound only holds for gaussian decay, not subgaussian. e.g. Rademachers are subgaussian, but gamma_2 is not a lower bound for Rademachers (example: gamma_2 of the ell_1 ball in p dimensions is $\Theta(\sqrt{\log p})$, but the Rademacher width equals $1$). * Lines 210-212 claim that (5) holds for any subexponential r.v.'s. This is again false for the same reason as the last bullet (also see my last comment below from the supplementary file).

FROM THE SUPPLEMENTARY FILE: * In line 124, "lose" should be "loose". Also, "For e.g." is redundant; it should just be "e.g." without the "for". * Equation (16) is slightly confusing. The "K" here is an absolute constant and should not be confused with the K from Lemma 1 (it seems the K from Lemma 1 is assumed to equal 1 here). * Below line 215, it states "the result is applicable to any process that has concentration inequality (16)". This is false. The upper bound of (17) is indeed true as long as (16) holds, but the lower bound is only true for the case when the entries of X are exactly exponential (as opposed to subexponential). For example, gaussians are subexponential, but the lower bound is wrong if the entries of X are actually gaussians (the gamma_1 term shouldn't be there). For a more trivial example, the distribution that is supported only on the number "zero" is subexponential, and the lower bound is obviously false for it (although I guess this trivial example doesn't quite work if you assume all variances are 1).
Summary: Technically, I found the jump from analyzing the gaussian case in previous work to the subexponential case in this submission to be quite a small jump. The new main theorem, Theorem 1, follows essentially immediately from the two theorems cited in Talagrand's book.

In any case, not having a big technical contribution is perfectly OK. More importantly, I think the submission could benefit greatly from more time spent on motivation in the introduction. See detailed comments below.

Author Feedback
Author rebuttal: We thank all reviewers for their comments and feedback.

Regarding motivating the need for sub-exponential noise and covariates: we will follow reviewers suggestions of expanding upon our terse motivation in the introduction. While such a generalization is certainly of mathematical and theoretical interest, we believe that this work will be of great practical interest as well: in varied domains as finance, insurance, climate, ecology, social network analysis etc. where measurement matrices with heavy tails and few samples are frequently encountered. We will add many examples to the final version, but as one concrete example here, we refer to the following works:
http://arxiv.org/ftp/arxiv/papers/1206/1206.4685.pdf
http://www.niculescu-mizil.org/papers/KDD09Climate-final.pdf
The covariates are from extreme value distributions, which are log-concave and hence sub-exponential. Our work could thus bridge the gap between statistical theory and practice in such applications.

Regarding technical contributions of our work:
1. We agree that Theorem 1 on the relation between the Gaussian and exponential width is simple. But we note that prior literature on this topic by Adamczak et al. etc. have more complex arguments even for the specific case of l1 penalization. Other work by Mendelson et al. introduces exponential widths but does not detail methods for bounding them. Our result is immediately applicable to a wide variety of norms for which gaussian widths have been bounded.
2. We note moreover, that the relationship between exponential and Gaussian widths is only one of many of our contributions: a second major contribution of our paper is the restricted eigenvalue (RE) condition result based on the VC dimensions argument. The RE condition is satisfied for heavy tailed design matrices with the same sample complexity as sub-Gaussians. This result was stated and proved in Lecue and Mendelson. But they prove results only for the l1 norm in the noiseless setting and they do not make connections with the Gaussian width. In contrast, all our results are in the noisy setting. We adapt their arguments for the error cone, as a consequence of which we are able to extend such arguments even to group sparse norms. We intend to explore in future work if this is true for all norms. On another larger point, we hope to enlighten readers regarding the gap between RE condition and the restricted isometry property (RIP). From Theorem 3 and Adamczak et al., for sub-exponential designs for l1 and group sparse norms, RIP has an additional (log p) factor sample complexity bound.

Regarding experiments: We will add figures that we subsequently plotted comparing sub-Gaussians and sub-exponentials using normalized sample sizes. In the noiseless case, only RE needs to be satisfied when the true vector is recovered with similar sample sizes in both cases. In the noisy setting, \|\theta^* - \hat{\theta}\|_2 decays slower for sub-exponential.

Assigned_reviewer_1:
1. We do believe that this work can be extended to the non-isotropic case to get (log p) worse sample complexity bounds for the RE condition. But we will like to explore if sharper bounds are possible in future work.
2. We will compare/contrast with Vershynin's work in the final version.
3. We thank the reviewer for feedback on the experiments. We will address it in the final version.

Assigned_reviewer_2:
We thank the reviewer for pointing out technical and stylistic changes to make in the writing. We will address these in the final version.

Assigned_reviewer_3:
The argument for the RE condition based on VC dimensions is indeed surprising and it remains to be seen if similar results are applicable to all norms.
We thank the reviewer for feedback on using normalized sample sizes for the experiments. We agree and will incorporate it in the final version.

Assigned_reviewer_4:
The values of the constants change from line to line. We will clarify this by notational changes in the final version.

Assigned_reviewer_5:
1. We acknowledge the comments given by the reviewer on our description on sub-exponential distributions and will make appropriate changes in the final version.
2. Our analysis for the regularized problem is based on the analysis in Banerjee et al., Negahban et al. where they have assumed \lambda_n satisfies the inequality in line 108. As they note there, this in some sense expresses a bound on the noise level.
3. We agree that the statement in Line 199-201, 210-212 in the main paper and line 215 in the supplement holds only for the upper bound. We will make this correction in the final version.
4. We will clarify issues regarding Equation (16) in the supplement in the final version.
5. We thank the reviewer for pointing out other technical and stylistic changes to make in the writing. We will address these in the final version.

Assigned_reviewer_6:
We thank the reviewer for their encouraging comments.